# Pupillometry evaluation of melanopsin retinal ganglion cell function and sleep-wake activity in pre-symptomatic Alzheimer's disease

**Angela J. Oh**[1]*, **Giulia Amore**[1,2], **William Sultan**[1], **Samuel Asanad**[1], **Jason C. Park**[3], **Martina Romagnoli**[2], **Chiara La Morgia**[2,4], **Rustum Karanjia**[1,5], **Michael G. Harrington**[6], **Alfredo A. Sadun**[1]

**1** Doheny Eye institute, UCLA Stein Eye Institute, University of California, Los Angeles, Department of Ophthalmology, Los Angeles, California, United States of America, **2** IRCCS Istituto delle Scienze Neurologiche di Bologna, UOC Clinica Neurologica, Bologna, Italy, **3** Columbia University, Department of Psychology, New York, New York, United States of America, **4** Dipartimento di Scienze Biomediche e Neuromotorie, Università di Bologna, Bologna, Italy, **5** University of Ottawa Eye Institute, Department of Ophthalmology, Ottawa, Ontario, Canada, **6** The Huntington Medical Research Institutes and Molecular Neurology Program, Pasadena, California, United States of America

* angelaoh@mednet.ucla.edu

**Data Availability Statement:** All relevant data are within the manuscript.

## Abstract

### Background

Melanopsin-expressing retinal ganglion cells (mRGCs), intrinsically photosensitive RGCs, mediate the light-based pupil response and the light entrainment of the body's circadian rhythms through their connection to the pretectal nucleus and hypothalamus, respectively. Increased awareness of circadian rhythm dysfunction in neurological conditions including Alzheimer's disease (AD), has led to a wave of research focusing on the role of mRGCs in these diseases. Postmortem retinal analyses in AD patients demonstrated a significant loss of mRGCs, and *in vivo* measurements of mRGC function with chromatic pupillometry may be a potential biomarker for early diagnosis and progression of AD.

### Methods

We performed a prospective case-control study in 20 cognitively healthy study participants: 10 individuals with pre-symptomatic AD pathology (pre-AD), identified by the presence of abnormal levels of amyloid $\beta_{42}$ and total Tau proteins in the cerebrospinal fluid, and 10 age-matched controls with normal CSF amyloid $\beta_{42}$ and Tau levels. To evaluate mRGC function, we used a standardized protocol of chromatic pupillometry on a Ganzfeld system using red (640 nm) and blue (450 nm) light stimuli and measured the pupillary light response (PLR). Non-invasive wrist actigraphy and standardized sleep questionnaires were also completed to evaluate rest-activity circadian rhythm.

### Results

Our results did not demonstrate a significant difference of the PLR between pre-AD and controls but showed a variability of the PLR in the pre-AD group compared with controls on

**Funding:** The authors received no specific funding for this work.

**Competing interests:** The authors have declared that no competing interests exist.

chromatic pupillometry. Wrist actigraphy showed variable sleep-wake patterns and irregular circadian rhythms in the pre-AD group compared with controls.

## Conclusions

The variability seen in measurements of mRGC function and sleep-wake cycle in the pre-AD group suggests that mRGC dysfunction occurs in the pre-symptomatic AD stages, preceding cognitive decline. Future longitudinal studies following progression of these participants can help in elucidating the relationship between mRGCs and circadian rhythm dysfunction in AD.

## Introduction

Alzheimer's disease (AD) is the most frequent type of dementia, characterized by abnormal accumulation of misfolded amyloid-ß (Aß) protein and hyperphosphorylated-Tau in the brain, in the form of amyloid plaques and neurofibrillary tangles respectively. Diagnosis of definite AD requires post mortem brain analysis, and thus is difficult to predict especially prior to clinical manifestations [1–3]. *In vivo* quantification of Aß and Tau proteins and their ratio in the cerebrospinal fluid (CSF), coupled with neuroimaging and neuropsychological studies, is often used to evaluate the likelihood of probable or possible AD diagnosis [4]. CSF biomarkers can also be helpful to identify asymptomatic patients at risk to develop AD in the future. The search for additional *in vivo* markers to improve early diagnosis of AD or measure progression is expanding and the eye represents a promising field of research [5].

Primary symptoms of AD include cognitive impairment and memory loss. In addition, visual complaints are reported by AD patients even in early stages of disease, usually reflecting the progressive neurodegeneration along the parietal-occipital cortex and the posterior visual pathways [5, 6]. Nevertheless, a more anterior involvement, of both the retina and optic nerve, is nowadays well established in AD and may precede brain pathology [4–6]. In human post-mortem specimens and animal models, histological studies demonstrated the characteristic Aß deposits and Tauopathy in the inner retina, which are associated with degeneration of retinal ganglion cells (RGCs) and macular and optic nerve thinning with predominant loss of larger fibers in the superior quadrant [7–9].

Many optical coherence tomography (OCT) studies confirmed *in vivo* the reduction of retinal nerve fiber layer (RNFL) thickness in AD patients compared to controls, with evidence of intermediate values in patients with mild cognitive impairment (MCI) [10, 11]. RNFL thinning, consistent with histological evidence, was seen in the superior and inferior quadrants with a relative sparing of the temporal quadrant [7]. This pattern differs from studies in patients with mitochondrial optic neuropathies and other neurodegenerative diseases, like Parkinson's and Huntington's Diseases, where there is a preferential involvement of the temporal quadrants due to loss of the smaller RCGs [7, 12].

Moreover, La Morgia and coauthors demonstrated a significant loss of intrinsically photosensitive RGC containing melanopsin (mRGCs) in AD compared to controls and that these cells are selectively affected by the amyloid pathology in post-mortem retinal tissues. MRGCs are part of a non-visual pathway responsible for the PLR and the light entrainment of circadian rhythms with multiple connections in the brain including the suprachiasmatic nucleus of the hypothalamus [13, 14]. Hannibal and coauthors demonstrated in humans the projections of mRGCs to the the suprachiasmatic nucleus, the site of circadian regulation *[15]*. Thus, mRGC

degeneration may explain the dysregulation of sleep-wake cycle characteristically observed in aging and Alzheimer's disease. This is supported by several actigraphy studies showing different profiles of circadian dysfunction with decreased sleep latency and overall sleep time compared to controls [7–9, 16–18].

Interest on the *in vivo* evaluation of mRGC function as a potential biomarker for different pathological conditions is growing. Melanopsin is a photopigment contained in mRGCs that selectively reacts to short-wavelength blue light (459–483 nm) [8, 9, 19, 20]. The immediate and apparent reaction to mRCGs depolarization in response to light is a pupillary constriction that can be assessed by chromatic pupillometry [8, 9, 19, 20]. Protocols using blue and red light flashes of increasing luminance, after dark or light adaptation, showed that mRGC-mediated pupillary response differs from cone or rod-driven responses [21, 22].The mRGC-mediated pupillary light response (PLR) shows a sustained response to colored stimuli, with pupil size returning slowly to baseline pupil amplitudes [21].

Chromatic pupillometry has been used to assess mRGC function in various conditions. Studies in Leber's hereditary optic neuropathy (LHON) demonstrated a selective preservation of mRGCs with a normal PLR [20]. While in glaucoma and Parkinson's disease, PLR amplitude was reduced compared with controls [9, 19, 23]. A few studies have used pupillometry to study mRGC function in AD [8, 24]. Pupillometry may also be helpful to measure early changes in mRGC function in older participants without clinical symptoms but with clinically probable AD differentiated by CSF or imaging biomarkers. However, the results have been variable and the utility of pupillometry in AD and pre-symptomatic AD remains inconclusive [24–28].

The purpose of this study was to specifically evaluate mRGC function with chromatic pupillometry in pre-symptomatic AD. Prolonged wrist actigraphy recording and sleep questionnaires were also completed to detect early changes in circadian sleep-wake cycle.

## Materials and methods

### Study participants

This was a prospective case-control study conducted at the Doheny Eye Center, Division of Neuro-ophthalmology, in Pasadena, California and the Huntington Medical Research Institutes. The study was approved by the Institutional Review Board of the Huntington Hospital, Pasadena, California and the University of California, Los Angeles (IRB#17–001645), as per their policies. The study was in compliance with the Declaration of Helsinki. Study participants over 60 years of age were recruited prospectively for this Brain Aging research study (Protocol 37937) from the local San Gabriel Valley area as described [4]. Only participants who were determined to have no cognitive impairment by Uniform Data Set-3 assessments as described in the National Alzheimer's Coordinating Center and consensus clinical conferencing, were included [29]. All participants provided written, informed consent of the study after the purpose and methods of the study were thoroughly explained. We classified these cognitively healthy individuals into two groups depending on the result of CSF analysis for Alzheimer's disease (AD) biomarkers: those with a pathological value of $A\beta_{42}$/Tau ratio were considered as pre-symptomatic AD (pre-AD) [4], or those with a normal $A\beta_{42}$/Tau ratio as a control group. This cutoff for the CSF $A\beta_{42}$/Tau ratio had been previously found to correctly classify >85% of individuals with clinically probable AD [4].

All participants completed thorough ophthalmic and cognitive evaluation and exhibited no significant differences in age, gender, education, medications, vascular risk factors, and magnetic resonance imaging features of small vessel disease. Exclusion criteria included were best corrected visual acuity less than 20/50, pre-existing macular pathologies, and neurological,

psychiatric, or ocular diseases. Participants who were diagnosed by a physician with sleep apnea or used continuous positive airway pressure machines were excluded from wrist actigraphy analyses. Participants in the control group with significant sleep disturbances as evaluated by sleep questionnaires were also excluded in analyses of wrist actigraphy and subjective sleep.

## Chromatic pupillometry evaluation

MRGC cell-mediated PLR was assessed using previously established chromatic pupillometry protocols [20, 21]. We focused on measuring the sustained response to the intense blue stimuli in the dark to target the melanopsin condition [21]. The dominant eye was tested, and the contralateral eye was patched for monocular testing. Participants sat with their eyes open in a dark room for 10 minutes to naturally dilate their eyes prior to chromatic pupillometry testing. Colored light stimuli were presented for one second each using a Ganzfeld ColorDome full-field stimulator (Diagnosys, UK, Ltd) with an integrated pupillometer to record the pupil response.

We referenced the protocol described by Park and coauthors, which describes isolating rod, cone, and melanopsin contributions to the PLR using different wavelengths, intensities, and adaptations [21]. In this study, we focused on the melanopsin condition of their protocol using intense photopically matched red and blue stimuli in the dark to specifically assess the contribution of mRGCs to the PLR [21]. We presented a flash stimulus of intense red light (620 nm) for 1 second at a luminance of 250 candels per square meter (2.3 log cd/m$^2$). We then repeated the recording with the photopically matched blue stimulus (450 nm) for 1 second. The inter-stimulus interval (ISI) was 20 seconds for the red stimulus and 40 seconds for the blue stimulus. All recordings were completed in the same order with the red stimulus followed by the blue stimulus. PLR recordings were repeated three times for each colored stimulus and the recordings were averaged. Participants were instructed to try their best to keep their eyes open during the duration of the light stimuli as well as for 10 seconds following the stimuli. Participants who blinked frequently during the recordings were given another opportunity to repeat the measurements.

Each trial of PLR for all participants were inspected visually by a masked administrator to confirm for data quality and additional artifacts. The PLRs for each participant were normalized by the baseline pupil size, defined as the mean pupil size during the 1 second before stimulus onset. Normalized pupil size was calculated by absolute pupil size/baseine pupil size. The peak amplitude or peak normalized pupil size was defined as the pupil size at the point of greatest constriction or the minimium pupil size. Sustained response was defined as the normalized median pupil size in the time window between 6 to 8 seconds from flash onset. We compared the pupil size at baseline, peak amplitude, and sustained response between the pre-AD and controls for the PLR following both colored stimuli and compared the results between the two groups.

## Wrist actigraphy and sleep questionnaires

Participants were instructed to wear a non-invasive, lightweight, waterproof wrist watch with an embedded actigraphy device (ActigraphMotionwatch8, Camntech, Ltd) on either wrist. Because there were only two actigraphy devices available for all participants and participants took each device home, only a subset of participants were able to complete both the pupillometry and actigraphy. Participants were instructed to wear the wrist actigraphy device for at least 5 consecutive days and to keep a sleep diary that recorded their bed and rise time, daytime naps, or any time they removed the watch for any reason. Participants were advised on removing the watch during water activities such as showering or swimming, which might damage the device. Patients with less than 5 consecutive days of recorded data were excluded.

Actigraphy analyses were completed using TheMotionware software (Camtech, Ltd). We compared non-parametric measures of circadian stability including intra-daily variability (IV, degree of fragmentation of activity-rest periods, range: 0–2), interdaily stability (IS, degree of regularity, range 0–1) and relative amplitude (RA, (M10 –L5)/ (M10 + L5), range 0–1), most 10 average (M10, activity during most ten active hours), and least 5 average (L5, activity for least five active hours) [30]. Intra-daily variability (range 0–2) measured the frequency of transitions between rest and activity. Interdaily stability (range of 0–1) measured the strength of rhythm and degree of regularity in the activity-rest pattern. Relative amplitude measured the normalized difference between the most active 10 hour period (M10) and the least active 5 hour period (L5). Total sleep time was defined as the minutes of rest activity during a night period. Sleep efficiency was calculated by the percentage of total sleep time divided by time in bed. All measurements were averaged during the nights of sleep for each participant and compared between the pre-AD and control groups.

All participants also completed three standardized self-administered questionnaires (Epworth Sleepiness Scale (ESS), Pittsburgh Sleep Quality Index (PSQI) and Berlin Questionnaire to evaluate the possible occurrence of sleep disturbances. Participants with pathologic scores on these questionnaires were differentiated as those who scored above 10 on the ESS, above 5 for PSQI, and high risk on the Berlin questionnaire.

## Statistical analyses

Statistical analysis was completed using commercial spreadsheet software (Microsoft Excel; Microsoft Corporation, Redmond, WA). All pupillometry recordings were evaluated to ensure that the average response of the three recordings for each colored stimuli was appropriate. Pupillometry data were analyzed offline using Prism software (San Diego, CA) paralleling the protocol used by several different studies [28, 31]. Recordings with multiple eye blinks that contaminated the PLR pattern or curvature were rejected.

After determining variables were not normally distributed by the Kolmogorov-Smirnov test, a non-parametric Mann Whitney U test was used to compare pupillometry and actigraphy measurements between controls and pre-AD participants. A two-tailed Spearman correlation was performed to measure the relationship between $A\beta_{42}$/Tau ratios with the chromatic pupillometry, wrist actigraphy, and sleep questionnaire measurements for both controls and pre-AD participants.

## Results

### Demographics

We recruited a total of 20 subjects, 10 controls (mean age: 72.7 ± 7.9 years, 70% female) and 10 pre-AD (mean age: 75.7 ± 6.3 years, 90% female). Demographic data of the participants are seen in Table 1. There was no significant difference in age, BMI, or visual acuity between controls and pre-AD groups. The values of $A\beta_{42}$ /Tau ratio in the CSF were significantly decreased in the pre-AD (1.5 ± 0.4) compared with controls (4.3 ± 1.2; p = 0.0007). There was no significant correlation between $A\beta_{42}$ /Tau ratio and age for all participants (controls r: 0.22, p = 0.54; pre-AD: r: 0.25, p = 0.52, nonparametric Spearman correlation).

### Pupillary light response with chromatic pupillometry

All 20 participants (10 pre-AD individuals and 10 age-matched controls) completed chromatic pupillometry recordings for both the red and blue stimuli. One control participant had a

**Table 1. Demographic data of pre-AD disease participants and controls.**

|  | Control | Pre-AD | P value |
|---|---|---|---|
| **Demographics** |  |  |  |
| Participants (n) | 10 | 10 |  |
| Female, % | 70 | 90 |  |
| Age, yrs | 72.7 ± 7.9 | 75.7 ± 6.3 | 0.65 |
| Body Mass Index | 28.5 ± 6.5 | 26.2 ± 2.9 | 0.62 |
| Visual Acuity, log MAR | 0.0 ± 0.0 | 0.1 ± 0.1 | 0.99 |
| $A\beta_{42}$ /Tau ratio | 4.3 ± 1.2 | 1.5 ± 0.4 | 0.0007 |

Mean ± standard deviation.

P-values were calculated using the non-parametric Mann-Whitney U Test.

Statistical significance was defined as P values < 0.05.

Pre-AD, pre-symptomatic Alzheimer's disease.

recording that did not resemble a PLR for the blue stimulus, and this recording was excluded from analysis.

All pre-AD participants and controls showed measurable PLRs with the intense red and blue stimuli (Fig 1). The blue light evoked a much stronger PLR with a longer sustained response compared to the red stimulus when matched for photopic luminance. The PLRs for the intense blue stimulus showed a slower return to baseline pupil size following light offset (blue lines in Fig 1) while the PLRs to the intense red stimulus showed a quick return to baseline pupil size (red lines in Fig 1). On average, the PLRs for both the red and blue stimuli were similar between the pre-AD and control group (Fig 1, solid red and blue lines). For the red stimulus, individual pre-AD PLRs (Fig 1A, dashed red lines) were similar to control PLRs (Fig 1B, dashed blue lines). However for the blue stimulus, there was PLR variability in the pre-AD group (Fig 1C, dashed blue lines) that was not seen in controls (Fig 1D, dashed blue lines). In the control group, all but one participant showed PLRs to the blue stimulus that were similar to each other (Fig 1D).

Quantitative measurements of PLR, including the baseline pupil size, normalized peak pupil size and sustained response, were not significantly different in pre-AD participants compared with controls for both the intense red and blue stimuli (Fig 2, Table 2). The average baseline pupil size was smaller in the pre-AD group compared to controls, although this was not statistically significant (Table 2). The difference between PLRs to the two colored stimuli were analyzed but neither peak nor sustained response showed a statistically significant difference between the pre-AD and control group.

$A\beta_{42}$ /Tau ratios were not significantly correlated in controls with the peak pupil size (red, r: 0.18, p = 0.63; blue: r: -0.43, p = 0.25) and sustained response (red, r: 0.18, p = 0.63; blue: r: -0.05, p = 0.91). There was no relationship for the pre-AD group as well, for the peak pupil size (red, r: -0.28, p = 0.42; blue: r: -0.30, p = 0.41) and sustained response (red, r: 0.22, p = 0.54; blue: r: -0.31, p = 0.39). Age was not significantly correlated with the peak pupil size and sustained response.

## Wrist actigraphy and sleep-wake activity

A subset of 12 participants, 5 controls and 7 pre-AD participants completed the wrist actigraphy recording. Compared with controls, pre-AD participants showed no statistically significant difference in sleep efficiency (Fig 3A, Table 3). Measurements of most 10 average (M10) for the wake period, and least 5 average (L5) for the night period were slightly decreased

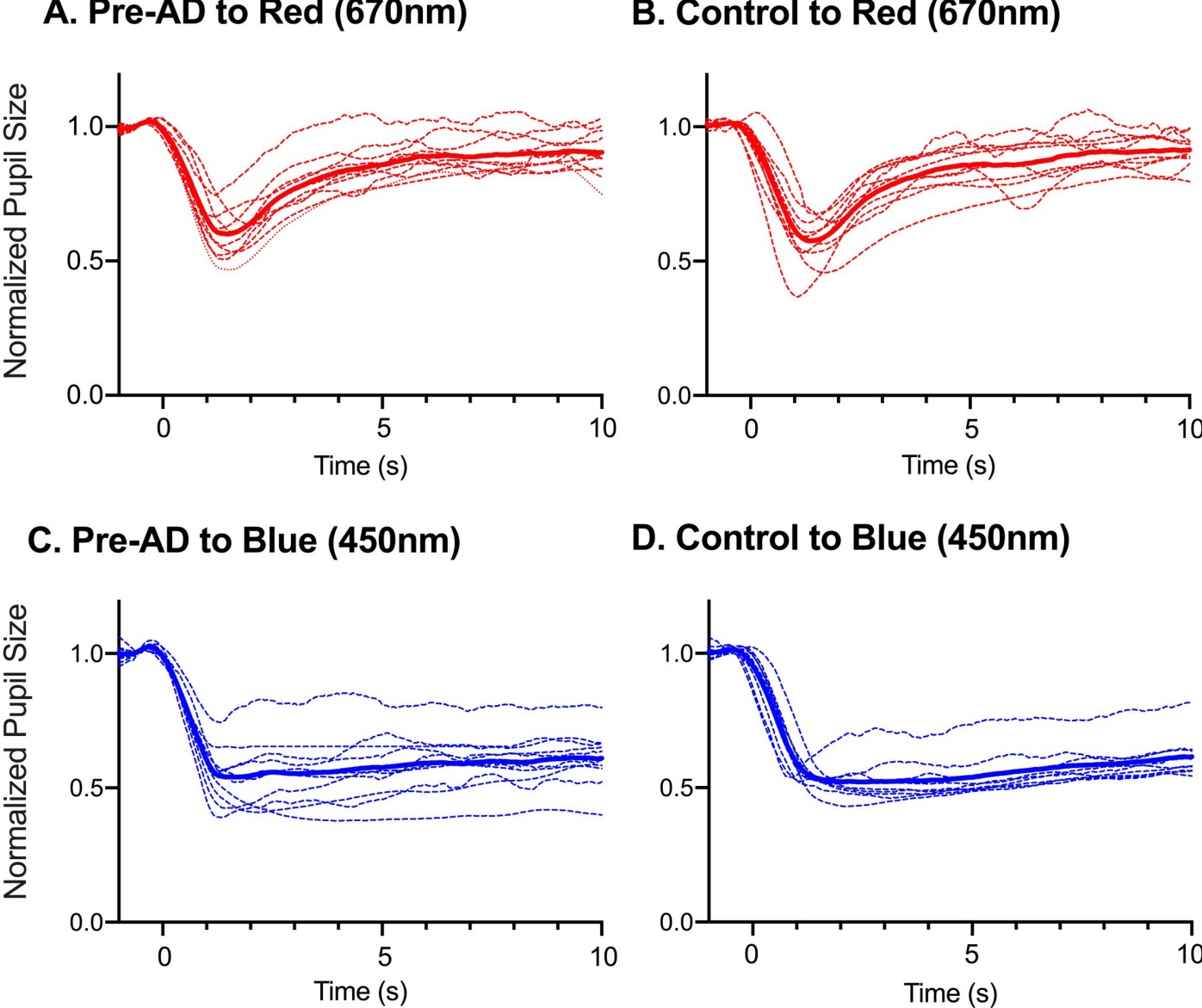

**Fig 1. Normalized pupillary light response (PLR) following photopically matched red (620 nm) and blue stimuli (450nm) at 250 or 2.3 log cd/m².** (A) Following the red stimulus, individual normalized PLRs of pre-AD individuals (n = 10, dashed red) and the average pre-AD PLR (solid red). (B) Individual normalized PLRs in controls (n = 10, dashed red) and the average control PLR (solid red). (C) Following the blue stimulus, individual normalized PLRs of pre-AD individuals (n = 10, dashed blue) and average pre-AD PLR (solid blue). (D) Individual controls (n = 9, dashed blue) and the average control PLR (solid blue). Pre-AD, pre-symptomatic Alzheimer's disease.

compared with controls (Table 3). However, these results were not statistically significant. Additional actigraphy measurements of sleep time in hours, relative amplitude (RA), intra-daily variability (IV), and interdaily stability (IS), showed no statistically significant difference in pre-AD participants compared with controls (Fig 3B, Table 3). In the control group, there was no significant correlation between Aß$_{42}$/Tau ratios for sleep parameters (actual sleep time, sleep efficiency, RA, IV, and IS). For the pre-AD group, there was a significant relationship between pathologic CSF marker ratios and IV, with greater Aß$_{42}$/Tau ratios correlating with increased IV ($r^2$: 0.88, p = 0.02). No other actigraphy markers were significantly correlated with Aß$_{42}$/Tau ratios in the pre-AD group.

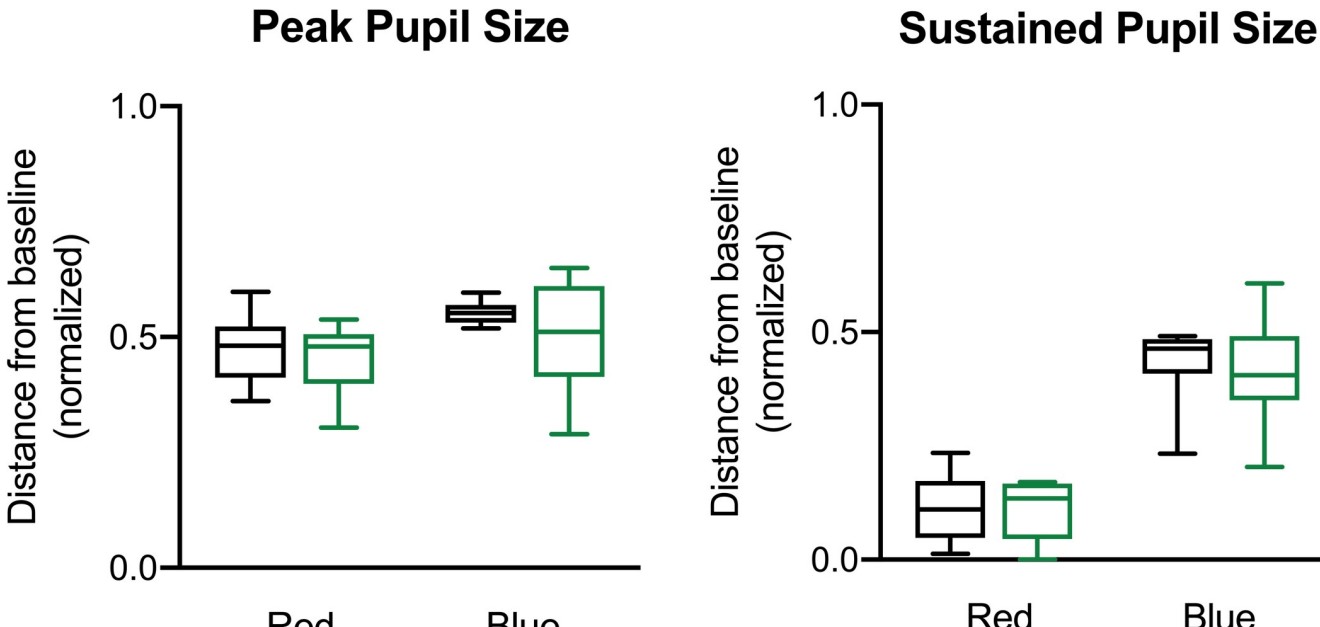

**Fig 2. Pupillometry measurements in pre-AD and controls.** Box whisker plots are shown comparing the pupillary light response following the red (620 nm) and blue stimulus (450 nm) for controls (black) and pre-AD participants (green). (A) Normalized peak pupil size and (B) normalized sustained pupil response as a change from baseline pupil size. Peak pupil size was defined as normalized pupil size at point of maximum pupil constriction or minimum pupil size. Sustained response was defined as the median normalized pupil size in the time window between 6 to 8 seconds from flash onset. Pre-AD, pre-symptomatic Alzheimer's disease.

In the pre-AD participants, actigraphy recordings showed variability with some participants resembling the pattern of controls (Fig 4A and 4B) and others with greater disruptions in rest-activity rhythms (Fig 4C). This pre-AD example (Fig 4C) spent less total sleep time with greater intra-daily variability and lower interdaily stability compared to the control example (Fig 4A) as well as the pre-AD participant that resembled the control (Fig 4B).

### Measurements of subjective sleep

All 20 participants completed all three sleep questionnaires. 4 participants (2 controls, 2 pre-AD) had pathologic ESS scores and 8 participants (5 controls, 3 pre-AD) had pathologic PSQI scores. None of the participants scored high risk for sleep apnea as evaluated by the Berlin Questionnaire. Three participants (1 control, 2 pre-AD) were either previously diagnosed with obstructive sleep apnea or were on treatment at the time of the study and were excluded from the analysis. One control participant was excluded based on their elevated PSQI score of 19 and ESS scores of 15.

We compared subjective sleep scores in a total of 16 participants, 8 controls and 8 pre-AD participants. Compared with controls, pre-AD participants did not report significantly different subjective scores of sleep as evaluated by all three sleep questionnaires (Table 3). There was also no statistically significant correlation between Aβ$_{42}$/Tau ratios and subjective sleep scores in all participants.

### Discussion

Melanopsin RGCs are affected in AD, but the functional impact of their loss is not well characterized. Our results measuring the function of mRGCs with chromatic pupillometry and rest-

**Table 2. Chromatic pupillometry data for controls and pre-AD.**

|  | Control | Pre-AD | P value |
|---|---|---|---|
| Red (620nm) |  |  |  |
| Participants | 10 | 10 |  |
| Baseline Pupil Size | 5.66 ± 1.1 | 4.81 ± 1.54 | 0.35 |
|  | (3.94–7.05) | (2.38–6.82) |  |
| Peak Pupil Size | 0.52 ± 0.07 | 0.54 ± 0.07 | 0.63 |
|  | (0.40–0.64) | (0.46–0.70) |  |
| Sustained Response | 0.89 ± 0.07 | 0.89 ± 0.07 | 0.74 |
|  | (0.77–0.99) | (0.83–1.03) |  |
| Blue (450nm) |  |  |  |
| Participants | 9 | 10 |  |
| Baseline Pupil Size | 5.15 ± 1.07 | 4.62 ± 1.46 | 0.50 |
|  | (3.94–6.89) | (2.34–6.68) |  |
| Peak Pupil Size | 0.45 ± 0.02 | 0.50 ± 0.12 | 0.55 |
|  | (0.40–0.48) | (0.35–0.71) |  |
| Sustained Response | 0.57 ± 0.08 | 0.59 ± 0.11 | 0.50 |
|  | (0.51–0.77) | (0.39–0.80) |  |

Mean ± standard deviation. Red (620nm) and blue stimulus (450 nm) presented at 250 cd/m2. Baseline pupil size = average pupil size during the 1 second before stimulus onset. Peak pupil size = normalized pupil size at point of maximum pupil constriction or minimum pupil size. Sustained response = normalized median pupil size in the time window between 6 to 8 seconds from flash onset. Statistical significance was defined as P values < 0.05. Pre-AD: Pre-symptomatic Alzheimer's disease.

activity circadian rhythm with actigraphy did not demonstrate statistically significant differences in the pupil response to colored stimuli and in quantitative sleep and circadian markers assessed by wrist actigraphy in pre-AD participants compared with controls. However, in the pre-AD cohort there is a higher variability in melanopsin function as evaluated by PLR to the blue stimulus. In addition, the variability of actigraphy results in pre-AD compared to controls also suggest potential changes in melanopsin function even in the very early stages of the pathology, before symptoms become apparent.

When assessing changes in melanopsin RGCs in disease states, identifying mRGC subtypes is important for understanding mRGC functions since each subtype likely has different non-imaging forming functions in the eye, including photo-synchronization of our circadian rhythm, sleep-wake cycle, pupillary light response, and cognition [32–35]. Six mRGC subtypes (M1 to M6) have been described in rodents [14, 36, 37]. In human retinas, Hannibal and coauthors identified six (M1, displaced M1, gigantic M1, gigantic displaced M1, M2, M4) mRGC subtypes [38]. The different subtypes are unevenly distributed in the human retinas, suggesting different roles and functions but their functions are not clearly elucidated [38]. One human study showed the projections of mRGCs to the to the suprachiasmatic nucleus (SCN), the center of our circadian clock [15], and one study demonstrated retinal projections to the SCN, the lateral geniculate complex including the pre-geniculate nucleus, the pretectal olivary nucleus, the nucleus of the optic tract, the brachium of the superior colliculus, and the superior colliculus in the macaque monkeys [39]. Post-mortem studies in aging populations above age 70 showed statistically significant age-dependent decrease in mRGC subtypes M1d and M3 cells, and trending changes in other mRGC subtypes [35, 40]. In one study in Parkinson's disease by

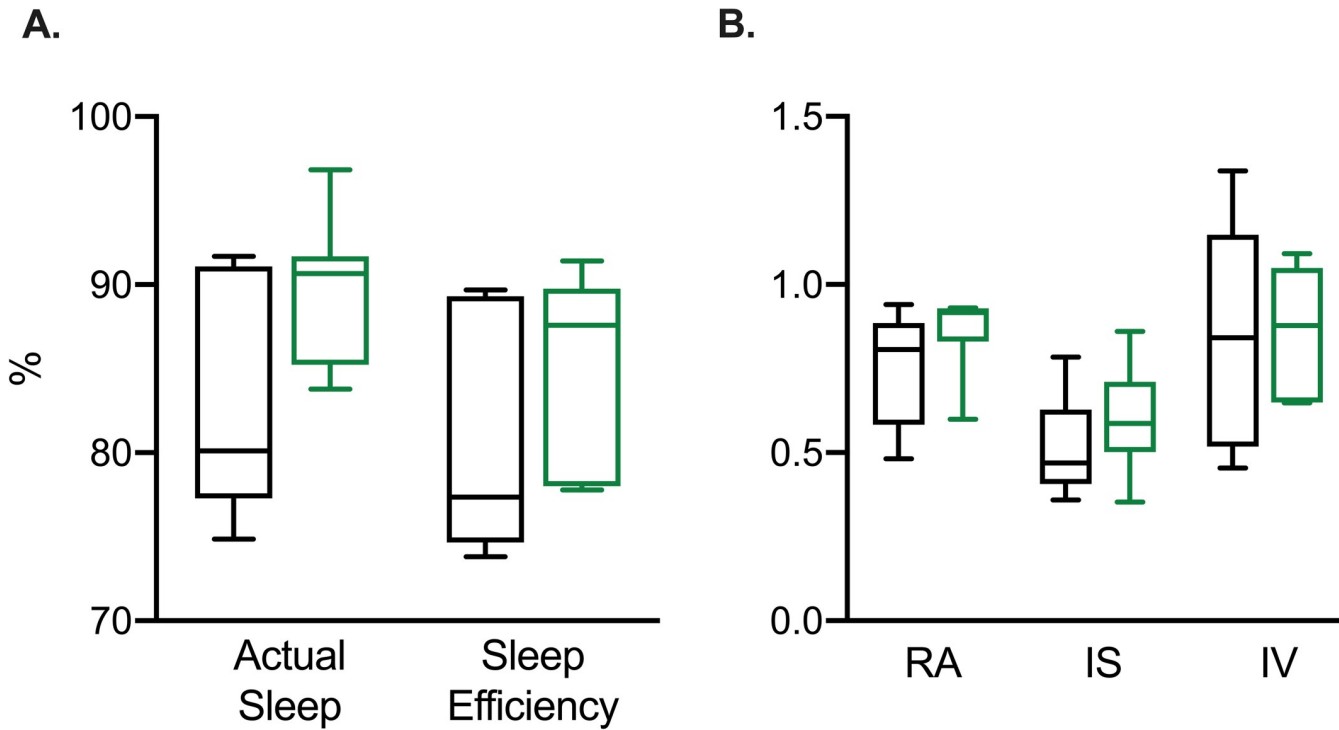

**Fig 3. Wrist actigraphy measurements in pre-AD and controls.** Box whisker plots of (A) Actual sleep (%) and Sleep efficiency (%), (B) Relative Amplitude (RA), Interdaily Stability (IS), Intra-daily Variability (IV) for controls (black, n = 5) and pre-AD individuals (green, n = 7). Actual sleep (%) was calculated as total sleep time in minutes divided by assumed sleep time (total elapsed time between 'Fell Asleep' and 'Woke Up' times). Sleep efficiency was calculated by the percentage of total sleep time divided by time in bed. Pre-AD, pre-symptomatic Alzheimer's disease.

Ortuño-Lizarán and colleagues, the M1d was the most affected mRGC subtype, showing a decrease in M1d density as well as morphological changes [41, 42].

Melanopsin subtypes in Alzheimer's retinas have not yet been characterized, and it is difficult to drive from chromatic pupillometry an inference on the specific subtype responsible of the phenotype without histological post-mortem data. Thus, the possible effect of pre-symptomatic AD on each mRGC subtype is unclear and yet to be explored. Based on our results (impairment of both pupillometry and actigraphic measures) and since these two functions are provided by different mRGC subtypes we cannot specifically identify which mRGC subtype is affected in pre-AD. Future post-mortem studies are needed to address this specific question.

A few pupillometry studies are available in AD and pre-AD using variable methods, flash intensities, and pupillometry measurements, and not specifically addressing the mRGC contribution [24–26]. One study used white flash stimuli in AD patients and noted characteristic changes in the maximum constriction velocity and maximum constriction acceleration [24]. Similarly, Frost and coauthors demonstrated similar significant changes in PLR in AD with an increase in latency and amount in PLR (constriction velocity and amplitude), and a more rapid return to baseline pupil size after offset compared to controls [26]. Van Stavern and coauthors measured the PLR following white light stimulus (949nm) in pre-AD individuals and found no significant differences in pupillometry evaluation compared with controls [25].

Chromatic pupillometry, with protocols focusing on the melanopsin condition with photopically matched intense blue and red stimuli, is thought to specifically measure mRGC function [20, 21]. By comparing the difference in responses to the red and blue stimuli, the

**Table 3. Wrist actigraphy and sleep quality data.**

| | Control | Pre-AD | P value |
|---|---|---|---|
| **Actigraphy** | | | |
| Participants | 5 | 7 | |
| Sleep Time, hr | 6.4 ± 1.3 | 6.7 ± 1.7 | 0.99 |
| Efficiency, % | 81.2 ± 7.6 | 85.1 ± 5.6 | 0.27 |
| Relative Amplitude | 0.75 ± 0.2 | 0.86 ± 0.1 | 0.34 |
| Intra-daily Variability | 0.83 ± 0.3 | 0.86 ± 0.2 | 0.76 |
| Interdaily Stability | 0.51 ± 0.2 | 0.59 ± 0.2 | 0.27 |
| Fragmentation Index | 28.6 ±14.3 | 31.5 ±14.0 | 0.99 |
| M10 | 11673 ± 6563 | 10724 ± 3290 | 0.88 |
| L5 | 1410 ± 854 | 753 ± 587 | 0.20 |
| **Sleep Questionnaires** | | | |
| Participants | 8 | 8 | |
| Epworth | 5.9 ± 4.1 | 4.1 ± 4.2 | 0.37 |
| Berlin Low, Berlin High, % | 75, 25 | 86, 14 | 0.99 |
| PSQI | 6.1 ± 2.1 | 6.1 ± 3.9 | 0.80 |
| Quality of sleep from PSQI | 3.6 ± 4.9 | 5.1 ± 6.8 | 0.73 |

Mean ± standard deviation.

Statistical significance was set at P value < 0.05.

Pre-AD, pre-symptomatic Alzheimer's disease. M10, Most 10 Average; L5, Least 5 Average; PSQI, Pittsburgh Sleep Quality Index.

melanopsin contribution can be isolated and measured. This has been successful in other patient groups, including LHON and retinitis pigmentosa (RP) [20, 21]. In one LHON patient, there was a significant delay in onset of mRGC-mediated PLR (blue stimulus) and no PLR to the photopically matched red stimuli. In RP, there was preservation of mRGC function with a sustained response to the blue stimulus that was larger than that of controls [20, 21]. In our study, both pre-AD and control participants showed pupil constriction after intense red and blue stimuli, suggesting preservation of cone and melanopsin function, respectively. The peak amplitude following both the red and blue stimuli was no different between the two groups. Although there was no statistically significant difference in the average PLR between the two groups, we found an increased variability of PLRs following the blue stimulus in pre-AD compared to controls, suggesting early changes in mRGC function. Overall, controls showed minimal variability in the PLR to the blue stimulus (Fig 1D), except in one participant. PLR in this participant showed a smaller sustained response compared to the average response. Interestingly, this participant scored pathologic scores of subjective sleep on both PSQI and ESS. It is possible, although without cognitive impairment, that changes in mRGC-driven PLR and rest-activity circadian changes display early mRGC dysfunction, signaling a possible progression to AD.

Comparing pupillometry results in pre-AD is challenging due to the different categorizations of pre-AD. In one study, pre-AD was labeled in healthy controls with high neocortical amyloid burden, and compared to controls finding a significant difference in maximum constriction velocity [26]. Van Stavern and coauthors classified pre-AD based on having one or both of the following biomarkers: pathologic CSF Aß$_{42}$ levels and abnormal mean cortical binding potential of Aß as measured on positron emission tomography (PET) [25]. Even using a different methodology and patient population, we did not find significant differences in

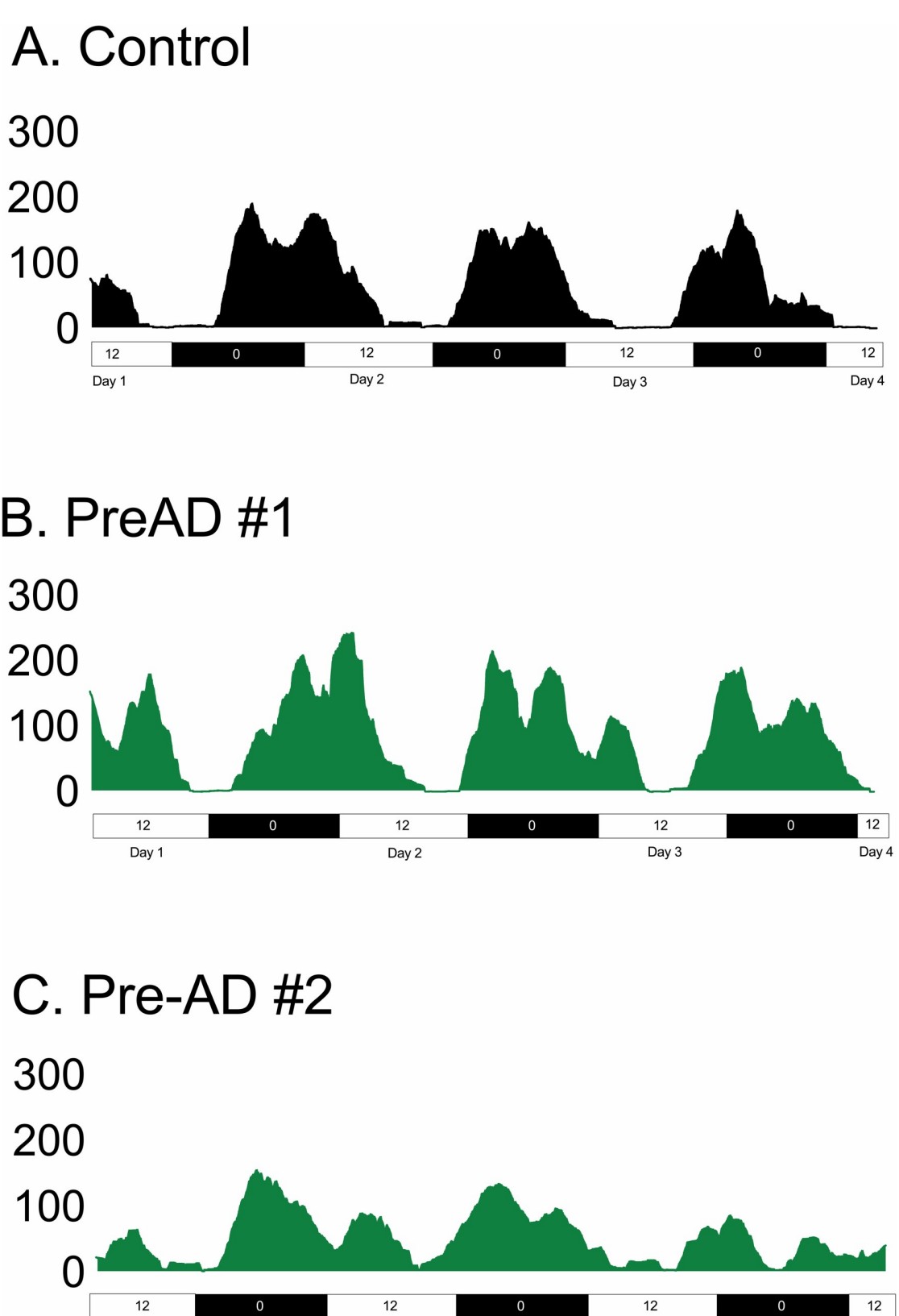

**Fig 4. Individual wrist actigraphy examples.** Actigraphy results in (A) a control example in black and (B) an example of pre-AD participants with an actigraphy profile similar to controls, and (C) another pre-AD participant who recorded greater irregularities in circadian rhythm and interruptions during sleep-wake activity in green, including a decrease in sleep time, lower sleep efficiency, greater intra-daily variability, and lower interdaily stability. Y-axis represents counts per epoch. Pre-AD, pre-symptomatic Alzheimer's disease.

quantitative measurements of PLR. We did, however, note an increased variability of PLR in the pre-AD group relative to controls, particularly with the intense blue stimulus.

Sleep-wake disturbances affect up to 40% of AD patients, impacting their quality of life, and disrupting their circadian rhythms [17, 35, 43]. Actigraphy recordings in AD patients have shown a significant decrease in sleep efficiency along with decreased activity during the day and increased activity at night, as measured on actigraphy [8]. In our study, assessment of circadian rhythm and rest-activity patterns in pre-AD failed to demonstrate significant differences compared to controls. There were no significant early changes in sleep efficiency in AD patients (75% in AD vs 90% in controls) [8]. Since the severity of dementia is positively correlated to the severity of circadian dysfunction, our results may be explained by the absence of cognitive dysfunction in our cohort, [16]. In another study, AD patients with poor stability of circadian parameters had the poorest prognosis, suggesting circadian dysfunction and irregular sleep-wake activity can predict cognitive outcome [44]. However, longitudinal studies have shown that actigraphy in AD does not worsen over one year, despite significant cognitive decline [44]. Further studies are needed to assess the presence and evolution of sleep disturbances in AD.

We also found a slight decrease in the average L5 and M10 actigraphy measurements in the pre-AD group, indicating a less restful or inactive period and less active and regular wake period, respectively. There was also a slight increase in intra-daily variability (IV) in pre-AD, and a significant positive correlation between IV and $A\beta_{42}$/Tau ratios in the CSF. This latter relationship may reflect a smaller total Tau level in the pre-AD participants in this early pathology stage. We also observed a higher variability in the circadian rhythm of the pre-AD group with some participants resembling the rhythm of controls, while others displayed irregularities in the rest-wake activity pattern. These results provide evidence of early changes in the circadian rhythm in pre-AD participants, also seen in another study, showing significantly worse sleep efficiency without changes in total sleep time in pre-AD [17]. In agreement with our quantitative findings, assessment of subjective sleep in our pre-AD group also showed no significant changes, similar to a report using the sleep questionnaires in AD patients [8]. Overall, sleep disturbances are also difficult to assess prior to cognitive changes, and larger samples and longitudinal studies are needed to further assess early changes in circadian dysfunction.

The major limitation of this study is the small sample size and cross-sectional nature of the study. We were also unable to follow the exact referenced protocol of wavelength and luminance to measure the melanopsin response with the blue stimuli. Park and coauthors assessed their melanopsin-mediated PLR using 470nm at a luminance of 450 cd/m$^2$ (2.5 log cd/m$^2$) while we used(450nm at 250 cd/m$^2$ or 2.3 log cd/m$^2$) [21]. Melanopsin-driven PLR is typically triggered at a luminance of 30 to 100 cd/m$^2$ (0.5–1 log cd/m$^2$), suggesting that our hardware was capable of sufficiently driving the PLR. In addition, we were unable to assess rest-activity circadian rhythm for all participants who completed pupillometry given the limited number of wrist actigraphy devices.

## Conclusions

The variability in the melanopsin driven light pupil response and in circadian rest-activity measures in our pre-AD cohort may suggest the presence of early mRGC pathology. Larger

and prospective studies are needed to further evaluate the role of objective chromatic pupillometry and circadian measurements as potential *in vivo* biomarkers for measuring melanopsin function. This can help clarify how melanopsin function is affected in the neuro-degeneration of AD pathology prior to cognitive changes.

## Author Contributions

**Conceptualization:** Angela J. Oh, Giulia Amore, William Sultan, Samuel Asanad, Chiara La Morgia, Rustum Karanjia, Michael G. Harrington, Alfredo A. Sadun.

**Data curation:** Angela J. Oh, Giulia Amore, William Sultan, Chiara La Morgia, Rustum Karanjia, Michael G. Harrington, Alfredo A. Sadun.

**Formal analysis:** Angela J. Oh, Giulia Amore, William Sultan, Jason C. Park, Rustum Karanjia, Michael G. Harrington, Alfredo A. Sadun.

**Funding acquisition:** Angela J. Oh, Samuel Asanad, Chiara La Morgia, Rustum Karanjia, Michael G. Harrington, Alfredo A. Sadun.

**Investigation:** Angela J. Oh, Giulia Amore, Chiara La Morgia, Rustum Karanjia, Michael G. Harrington, Alfredo A. Sadun.

**Methodology:** Angela J. Oh, Giulia Amore, Samuel Asanad, Jason C. Park, Martina Romagnoli, Chiara La Morgia, Rustum Karanjia, Michael G. Harrington, Alfredo A. Sadun.

**Project administration:** Angela J. Oh, Giulia Amore, Samuel Asanad, Chiara La Morgia, Rustum Karanjia, Michael G. Harrington, Alfredo A. Sadun.

**Resources:** Angela J. Oh, Giulia Amore, Samuel Asanad, Jason C. Park, Martina Romagnoli, Chiara La Morgia, Rustum Karanjia, Michael G. Harrington, Alfredo A. Sadun.

**Software:** Angela J. Oh, Giulia Amore, Jason C. Park, Martina Romagnoli, Chiara La Morgia, Rustum Karanjia, Michael G. Harrington, Alfredo A. Sadun.

**Supervision:** Angela J. Oh, Giulia Amore, Chiara La Morgia, Rustum Karanjia, Michael G. Harrington, Alfredo A. Sadun.

**Validation:** Angela J. Oh, Giulia Amore, Jason C. Park, Chiara La Morgia, Rustum Karanjia, Michael G. Harrington, Alfredo A. Sadun.

**Visualization:** Angela J. Oh, Giulia Amore, Chiara La Morgia, Rustum Karanjia, Michael G. Harrington, Alfredo A. Sadun.

**Writing – original draft:** Angela J. Oh, Giulia Amore.

**Writing – review & editing:** Angela J. Oh, Giulia Amore, William Sultan, Samuel Asanad, Jason C. Park, Martina Romagnoli, Chiara La Morgia, Rustum Karanjia, Michael G. Harrington, Alfredo A. Sadun.

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
