## [Decision Letter · Decision Letter 0]

23 Oct 2019

PONE-D-19-26357

Pupillometry evaluation of melanopsin retinal ganglion cell function and sleep-wake activity in pre-symptomatic Alzheimer’s disease

PLOS ONE

Dear Mrs. Oh,

Thank you for submitting your manuscript to PLOS ONE. After careful consideration, we feel that it has merit but does not fully meet PLOS ONE’s publication criteria as it currently stands. Therefore, we invite you to submit a revised version of the manuscript that addresses the points raised during the review process.

We would appreciate receiving your revised manuscript by Dec 07 2019 11:59PM. To enhance the reproducibility of your results, we recommend that if applicable you deposit your laboratory protocols in protocols.io, where a protocol can be assigned its own identifier (DOI) such that it can be cited independently in the future. For instructions see: http://journals.plos.org/plosone/s/submission-guidelines#loc-laboratory-protocols

We look forward to receiving your revised manuscript.

Kind regards,

Claudio Liguori

Academic Editor

PLOS ONE

**Journal Requirements:**

2. Thank you for stating that “The funders had no role in study design, data collection and analysis, decision to publish, or preparation of the manuscript” in your financial disclosure.

Please also provide the name of the funders of this study (as well as grant numbers if available) in your financial disclosure statement.

**Additional Editor Comments (if provided):**

Please improve the manuscript following the very minor comments of the Reviewers.

**Comments to the Author**

1. Is the manuscript technically sound, and do the data support the conclusions?

Reviewer #1: Yes

Reviewer #2: Yes

2. Has the statistical analysis been performed appropriately and rigorously? 

Reviewer #1: Yes

Reviewer #2: Yes

3. Have the authors made all data underlying the findings in their manuscript fully available?

Reviewer #1: Yes

Reviewer #2: Yes

4. Is the manuscript presented in an intelligible fashion and written in standard English?

Reviewer #1: Yes

Reviewer #2: Yes

5. Review Comments to the Author

Reviewer #1: Minor:

P. 4 l. 86: "...explain the dysregulation of the sleep-wake cycle characteristically observed in ". Something is missing in this sentences: AD?

please provide information on the authors contribution to the work

Reviewer #2: Pupillometry evaluation of melanopsin retinal ganglion cell function and sleep-wake activity in pre-symptomatic Alzheimer’s disease

Reviewer comments to author

A reduction in PLR in Alzheimer disease was previously described ( Tales et al., 2001; La Morgia et al., 2016). The paper by Dra. Angela J. Oh B.A. and colleagues provide evidences that the PLR and circadian rhythms are already affected in pre-symptomatic Alzheimer’s disease patients.

mRGCs decrease has been described in AD patients (La Morgia et al., 2016). The authors of this paper suggest an early mRGCs pathology before the symptoms of the disease appear, which is very interesting because using non-invasive techniques like PLR we could diagnose the disease before the symptoms manifest in the very early stages of the pathology.

Overall, the manuscript is well written and easy to follow. The experiments are convincing, the data presented is robust and the main findings are very interesting The figures are clear.

Only acomment below needs to be addressed.

Minor comments

A brief explanation of the different types of mRGCs and the possible affectation of each of these according to the results found is missing.

The different subtypes and the uneven distribution of melanopsin cells in the human retina seem to be associated with different NIF function.

Based on the results obtained in terms of PLR and circadian rhythms in AD, could you hypothesize any conclusions as to which type of mRGCs might be affected first or do you think that the loss of these cells occurs homogeneously?

6. PLOS authors have the option to publish the peer review history of their article (what does this mean?). If published, this will include your full peer review and any attached files.

Reviewer #1: Yes: Jens Hannibal

Reviewer #2: Yes: Gema Esquiva

---

## [Author Response · Author response to Decision Letter 0]

19 Nov 2019

Reviewer #1: Minor:

P. 4 l. 86: "...explain the dysregulation of the sleep-wake cycle characteristically observed in ". Something is missing in this sentences: AD?

Thank you for your comments. This has been edited and included under Introduction: Page 4 Line 89-90.

Please provide information on the authors contribution to the work

Author contributions are now included at the bottom of the manuscript under Author contributions: Page 23 Line 503 - 516.

Reviewer #2: 

A reduction in PLR in Alzheimer disease was previously described ( Tales et al., 2001; La Morgia et al., 2016). The paper by Dra. Angela J. Oh B.A. and colleagues provide evidences that the PLR and circadian rhythms are already affected in pre-symptomatic Alzheimer’s disease patients.mRGCs decrease has been described in AD patients (La Morgia et al., 2016). The authors of this paper suggest an early mRGCs pathology before the symptoms of the disease appear, which is very interesting because using non-invasive techniques like PLR we could diagnose the disease before the symptoms manifest in the very early stages of the pathology.Overall, the manuscript is well written and easy to follow. The experiments are convincing, the data presented is robust and the main findings are very interesting The figures are clear. Only a comment below needs to be addressed.

A brief explanation of the different types of mRGCs and the possible affectation of each of these according to the results found is missing.

Thank you for your comments. 

We completely agree that identifying mRGC subtypes is important for understanding mRGC functions since each subtype likely has different non-imaging forming functions in the eye, including photo-synchronization of our circadian rhythm, sleep-wake cycle, pupillary light response, and cognition [1-4]. Six mRGC subtypes (M1-M6) have been described in rodent studies [5-7]. In human retinas, Hannibal and coauthors identified described six (M1, displaced M1, gigantic M1, gigantic displaced M1, M2, M4) mRGC subtypes [8]. The different subtypes are unevenly distributed in the human retina, suggesting different roles and functions but their functions are not clearly elucidated in humans [8]. One human study showed the projections of mRGCs to the to the suprachiasmatic nucleus (SCN), the center of our circadian clock [9], and one study demonstrated retinal projections to the SCN, the lateral geniculate complex including the pre-geniculate nucleus, the pretectal olivary nucleus, the nucleus of the optic tract, the brachium of the superior colliculus, and the superior colliculus in the macaque monkeys [10]. Post-mortem studies in aging populations above age 70 showed statistically significant age-dependent decrease in mRGC subtypes M1d and M3 cells, and trending changes in other mRGC subtypes [4, 11]. In one study in Parkinson’s disease by Ortuño-Lizarán and colleagues, the M1d was the most affected mRGC subtype, showing a decrease in M1d density as well as morphological changes [12]. Melanopsin subtypes in Alzheimer’s retinas have not been characterized, and it is difficult to drive from chromatic pupillometry an inference on the specific subtype responsible of the phenotype without histological post-mortem data. Thus, the possible effect of pre-symptomatic AD on each mRGC subtype is unclear and yet to be explored. 

This concept has been added to the Discussion: Page 18, Line 379 – 397.

The different subtypes and the uneven distribution of melanopsin cells in the human retina seem to be associated with different NIF (non-imaging forming) function.

Based on the results obtained in terms of PLR and circadian rhythms in AD, could you hypothesize any conclusions as to which type of mRGCs might be affected first or do you think that the loss of these cells occurs homogeneously?

Based on our results (impairment of both pupillometry and actigraphic measures) and since these two functions are provided by different mRGC subtypes we cannot specifically identify which mRGC subtype is affected in pre-symptomatic AD. Future post-mortem studies are needed to address this specific question. 

This information has been added to the Discussion, Page 19, Line 398 – 405.

 

References 

1. Li JY, Schmidt TM. Divergent projection patterns of M1 ipRGC subtypes. J Comp Neurol. 2018;526(13):2010-8. Epub 2018/06/12. doi: 10.1002/cne.24469. PubMed PMID: 29888785; PubMed Central PMCID: PMCPMC6158116.

2. Baver SB, Pickard GE, Sollars PJ, Pickard GE. Two types of melanopsin retinal ganglion cell differentially innervate the hypothalamic suprachiasmatic nucleus and the olivary pretectal nucleus. Eur J Neurosci. 2008;27(7):1763-70. Epub 2008/03/29. doi: 10.1111/j.1460-9568.2008.06149.x. PubMed PMID: 18371076.

3. Chen SK, Badea TC, Hattar S. Photoentrainment and pupillary light reflex are mediated by distinct populations of ipRGCs. Nature. 2011;476(7358):92-5. Epub 2011/07/19. doi: 10.1038/nature10206. PubMed PMID: 21765429; PubMed Central PMCID: PMCPMC3150726.

4. Esquiva G, Hannibal J. Melanopsin-expressing retinal ganglion cells in aging and disease. Histol Histopathol. 2019:18138. Epub 2019/06/21. doi: 10.14670/HH-18-138. PubMed PMID: 31219170.

5. Schmidt TM, Chen SK, Hattar S. Intrinsically photosensitive retinal ganglion cells: many subtypes, diverse functions. Trends Neurosci. 2011;34(11):572-80. Epub 2011/08/06. doi: 10.1016/j.tins.2011.07.001. PubMed PMID: 21816493; PubMed Central PMCID: PMCPMC3200463.

6. Reifler AN, Chervenak AP, Dolikian ME, Benenati BA, Meyers BS, Demertzis ZD, et al. The rat retina has five types of ganglion-cell photoreceptors. Exp Eye Res. 2015;130:17-28. Epub 2014/12/03. doi: 10.1016/j.exer.2014.11.010. PubMed PMID: 25450063; PubMed Central PMCID: PMCPMC4276437.

7. Quattrochi LE, Stabio ME, Kim I, Ilardi MC, Michelle Fogerson P, Leyrer ML, et al. The M6 cell: A small-field bistratified photosensitive retinal ganglion cell. J Comp Neurol. 2019;527(1):297-311. Epub 2018/10/13. doi: 10.1002/cne.24556. PubMed PMID: 30311650; PubMed Central PMCID: PMCPMC6594700.

8. Hannibal J, Christiansen AT, Heegaard S, Fahrenkrug J, Kiilgaard JF. Melanopsin expressing human retinal ganglion cells: Subtypes, distribution, and intraretinal connectivity. J Comp Neurol. 2017;525(8):1934-61. Epub 2017/02/06. doi: 10.1002/cne.24181. PubMed PMID: 28160289.

9. Hannibal J, Hindersson P, Ostergaard J, Georg B, Heegaard S, Larsen PJ, et al. Melanopsin is expressed in PACAP-containing retinal ganglion cells of the human retinohypothalamic tract. Invest Ophthalmol Vis Sci. 2004;45(11):4202-9. Epub 2004/10/27. doi: 10.1167/iovs.04-0313. PubMed PMID: 15505076.

10. Hannibal J, Kankipati L, Strang CE, Peterson BB, Dacey D, Gamlin PD. Central projections of intrinsically photosensitive retinal ganglion cells in the macaque monkey. J Comp Neurol. 2014;522(10):2231-48. Epub 2014/04/23. doi: 10.1002/cne.23588. PubMed PMID: 24752373; PubMed Central PMCID: PMCPMC3996456.

11. Esquiva G, Lax P, Perez-Santonja JJ, Garcia-Fernandez JM, Cuenca N. Loss of Melanopsin-Expressing Ganglion Cell Subtypes and Dendritic Degeneration in the Aging Human Retina. Front Aging Neurosci. 2017;9:79. Epub 2017/04/20. doi: 10.3389/fnagi.2017.00079. PubMed PMID: 28420980; PubMed Central PMCID: PMCPMC5378720.

12. Lax P, Ortuno-Lizaran I, Maneu V, Vidal-Sanz M, Cuenca N. Photosensitive Melanopsin-Containing Retinal Ganglion Cells in Health and Disease: Implications for Circadian Rhythms. Int J Mol Sci. 2019;20(13). Epub 2019/07/03. doi: 10.3390/ijms20133164. PubMed PMID: 31261700.

---

## [Editor Report · Decision Letter 1]

22 Nov 2019

Pupillometry evaluation of melanopsin retinal ganglion cell function and sleep-wake activity in pre-symptomatic Alzheimer’s disease

PONE-D-19-26357R1

Dear Dr. Oh,

We are pleased to inform you that your manuscript has been judged scientifically suitable for publication and will be formally accepted for publication once it complies with all outstanding technical requirements.

With kind regards,

Claudio Liguori

Academic Editor

PLOS ONE

Additional Editor Comments (optional):

We thanks the Authors for addressing the reviewers' comments.
---

## [Editor Report · Acceptance letter]

3 Dec 2019

PONE-D-19-26357R1 

Pupillometry evaluation of melanopsin retinal ganglion cell function and sleep-wake activity in pre-symptomatic Alzheimer’s disease 

Dear Dr. Oh:

I am pleased to inform you that your manuscript has been deemed suitable for publication in PLOS ONE. Congratulations! Your manuscript is now with our production department. 

With kind regards,

on behalf of

Dr. Claudio Liguori 

Academic Editor

PLOS ONE